

# Merging holography, fluorescence, and machine learning for in situ, continuous characterization and classification of airborne microplastics

Nicholas D. Beres[1*], Julia Burkart[1**], Elias Graf[2], Yanick Zeder[2], Lea Ann Dailey[3], Bernadett Weinzierl[1]

[1] Faculty of Physics, Aerosol Physics and Environmental Physics, University of Vienna, Vienna, Austria

[2] Swisens AG, Emmen, Switzerland

[3] Department of Pharmaceutical Sciences, University of Vienna, Vienna, Austria

*   now at: Division of Atmospheric Sciences, Desert Research Institute, Reno, NV, USA

** now at: Sonnblick Observatory, Geosphere Austria, Vienna, Austria

Correspondence to: Nicholas D. Beres (nic.beres@dri.edu)

**Abstract.** The continued increase in global plastic production and poor waste management ensures that plastic pollution is a serious environmental concern for years to come. Because of their size, shape, and relatively low density, plastic particles between 1-1000 μm in size (known as microplastics, or MPs) emitted directly into the environment ("primary") or created due

to degradation ("secondary") may be transported through the atmosphere, similar to other coarse-mode particles, such as mineral dust. MPs can thus be advected over great distances, reaching even the most pristine and remote areas of the Earth, and may have significant negative consequences for humans and the environment. The detection and analysis of MPs once airborne, however, remains a challenge because most observational methods are offline and resource-intensive, and, therefore, are not capable of providing continuous quantitative information.

In this study, we present results using an online, in situ airflow cytometer (SwisensPoleno Jupiter; Swisens AG; Emmen, Switzerland) – coupled with machine learning – to detect, analyze, and classify airborne, single-particle MPs in near real time. The performance of the instrument to differentiate single-particle MPs of five common polymer types (including polypropylene, polyethylene, polyamide, poly(methyl methacrylate), and polyethylene terephthalate) was investigated under laboratory conditions using combined information about their size and shape (determined using holographic imaging) and

fluorescence measured using three excitation wavelengths and five emission detection windows. The classification capability using these methods was determined alongside other coarse-mode aerosol particles with similar morphology or fluorescence characteristics, such as a mineral dust and several pollen taxa.

     The tested MPs exhibit a measurable fluorescence signal that not only allows them to be distinguished from the other fluorescent particles, such as pollen, but can also be differentiated from each other, with high (> 90%) classification accuracy

based on their multispectral fluorescence signatures. The classification accuracies of machine learning models using only holographic images of particles, only the fluorescence response, and combined information from holography and fluorescence





to predict particle type are presented and compared. The results provide a foundation towards significantly improving the understanding of the properties and types of MPs present in the atmosphere.

## 1 Introduction

Plastics composed of synthetic or semi-synthetic polymer materials are ubiquitous in nearly all components of contemporary society. From packaging to consumer products to roadway materials, plastics are utilized because of their low cost of production and material properties. Due to these factors and rising demand, plastic production has been increasing by approximately 8.4% annually, where only ~9% of plastics are recycled, 12% are incinerated, and the rest accumulates in landfills and in the environment (Geyer et al., 2017). In recent years, public awareness and concern of plastic pollution as a

global environmental crisis is increasing (Davison et al., 2021), while, concurrently, the amount of plastic pollution in the environment has more than doubled in the period from 2000 to 2019 (Agrawala et al., 2022).

Plastic may be introduced into the environment through their origin as "primary" particles, i.e., purposefully manufactured particles for specific applications, such as personal care products (Fendall and Sewell, 2009) or industrial abrasives and paints (Verschoor et al., 2016). Once in the environment, plastics may undergo physical (e.g. mechanical),

radiative, chemical, and biological degradation, which alters their size, shape, and mobility within their environment (Othman et al., 2021; Brandon et al., 2016; Mao et al., 2020; Zhang et al., 2021). This degradation produces "secondary" fragments or particles. Primary or secondary particles are categorized into various size classes: macroplastics (> 1 cm), mesoplastics (between 1-10 mm), microplastics (referred throughout this publication as MPs; 1-1000 µm), and nanoplastics (1-1000 nm) (Hartmann et al., 2019). While MPs have been a known source of contamination in aquatic ecosystems (Cole et al., 2011;

Cózar et al., 2014; Barnes et al., 2009), an interest in research to better understand airborne particles has been on the rise (Brahney et al., 2021; Beaurepaire et al., 2021; Enyoh et al., 2019). Because of their size, shape, and material characteristics (such as their low density, (Driedger et al., 2015)), MPs and nanoplastics may be emitted into the atmosphere and transported long distances (Brahney et al., 2021), similar to other coarse-mode (maximum length > 1 µm) particles, such as mineral dust (Schepanski, 2018; Weinzierl et al., 2017), reaching even the most pristine and remote areas of the Earth (Brahney et al.,

2020a; Evangeliou et al., 2020; Bergmann et al., 2019; Aves et al., 2022). In addition, airborne MPs may cause significant health impacts if inhaled, as some particles can be in the respirable size range (Stuart, 1984; Gasperi et al., 2018), toxic (Prata et al., 2020), and bio-persistent (Mammo et al., 2020). Understanding the health impacts of microplastic particles is still evolving, and knowing their concentration, size distribution, and polymer type is imperative to address this growing concern (Prata, 2018).

The atmosphere remains the least understood environmental compartment for the fate of MPs (Akdogan and Guven, 2019; Zhang et al., 2020). The ubiquity of MPs in the environment and this lack of understanding has created the need for reliable, fast, and quantitative analysis methods. In particular, significant progress in studying the impact of atmospheric MPs is hindered by the lack of analytical methods which can effectively characterize particles in situ and in the size range relevant



to atmospheric transport. As particle size decreases, the time and effort required for identification of the plastic particles increases (Shim et al., 2017) and the size limits of detection for common, robust microplastic identification instruments are reached (such as 10-25 µm for FTIR and Raman spectroscopy) (Primpke et al., 2020). Additionally, most conventional methods of MP detection and/or characterization are offline (i.e., they do not measure continuously) and require tedious sample preparation (Primpke et al., 2020). Many standard analysis protocols are also limited in the information they can provide about the MPs. For example, some methods may be limited to providing only information connected to the chemical signature of material being analyzed, while others – such as the popular methods utilizing optical microscopy – may only provide limited information about MP size and relative abundance (Shim et al., 2017; Primpke et al., 2020).

An often-overlooked material property of airborne microplastics that has the potential to specify particle type is their natural ability to fluoresce, or autofluoresce, which results from the spontaneous emission of light at one wavelength by the fluorophores of the polymers from excited electromagnetic states when exposed to higher-energy, lower-wavelength light (Lakowicz, 2006). Some commercially available polymers have previously been examined for their autofluorescence (Allen et al., 1976; Lionetto et al., 2022; Asfour et al., 2020; Hawkins and Yager, 2003; Piruska et al., 2005; Spizzichino et al., 2016; Monteleone et al., 2021c, a; Könemann et al., 2018), but the identification of polymer types using their autofluorescence has been limited. Most studies examine *extrinsic* fluorescence of MPs, which is a method of applying a dye stain that adheres to the plastics (Primpke et al., 2020; Capolungo et al., 2021), which only provide a means to distinguish MPs from other non-fluorescing materials when viewed on filter media from an optical microscope (Maes et al., 2017; Erni-Cassola et al., 2017). This technique may be prone to misidentification (Beaurepaire et al., 2021), and, like other popular MP identification methods, is offline and labor intensive. No study has used the intrinsic fluorescence of polymers for airborne particle identification and characterization in situ. However, the recent introduction of the SwisensPoleno air-flow cytometer (Swisens AG; Emmen, Switzerland), which was shown to classify biological aerosol particles with high accuracy (Sauvageat et al., 2020; Erb et al., 2023a, b), provides an opportunity to explore the identification and characterization of other airborne particles, such as MPs. The SwisensPoleno (model: Jupiter) characterizes single particles by combining sensor information from digital holography from two orthogonal holographic imagers, and steady state spectrally resolved fluorescence intensity. The multi-method platform is complimented by state-of-the-art machine learning algorithms that provide the classification of airborne particle type in near real-time.

The objective of this study is to assess the fluorescence response of various common microplastics and to identify whether this information, together with holographic image of individual particles, measured using the SwisensPoleno can be used to distinguish MPs from other particle types. This fluorescence and other measured parameters, such as the particle morphology, are compared to data of other airborne, coarse-mode particles, including mineral dust, several taxa of pollen, and water droplets. These comparisons yield an estimation of viability for the online, in situ detection and classification of airborne MPs using the SwisensPoleno multi-sensor approach.



## 2 Methods

### 2.1 SwisensPoleno

The SwisensPoleno (Swisens AG; Emmen, Switzerland) is an air-flow cytometer providing continuous, in situ characterization of single, coarse-mode aerosol particles using multiple measurement methods in a single instrument. It combines sensor information to characterize single particles using digital holography from two orthogonal holographic imagers and spectrally resolved fluorescence intensity measurements. In addition, the instrument also provides a measurement of elastic forward- and polarized side-scattering of each particle; for this study, however, we focus only on using only the fluorescence and holographic imaging systems of the SwisensPoleno. The final component of this multi-method instrument is an integrated machine learning classification model, allowing the instrument to identify particle type in near-real-time by training models using all measured properties of individual airborne particles.

The SwisensPoleno resolves two digital holograms of the same single particle in the sample stream using digital in-line holography (Berg, 2022; Berg and Videen, 2011), with imaging sensors placed perpendicular to each other and the imaging plane perpendicular to the sample flow. Using only holographic imaging and image analysis, the SwisensPoleno has been used to detect and classify pollen particles from several different plant species in the size range between 10 – 200 μm with high accuracy (Sauvageat et al., 2020), later adapted to identify fungal spores during ambient monitoring in Switzerland (Erb et al., 2023a), and recently shown to be successful when combining holography and fluorescence information for pollen classification (Erb et al., 2023b). After hologram reconstruction and processing, each particle image is 200x200 pixels, with a resolution of 0.595 μm per pixel. For size and shape statistics, each holographic image is binarized and analyzed using the scikit-image software package (van der Walt et al., 2014) to determine a wide range of characteristic image properties (e.g., mean pixel intensity) and morphological features of each particle, including shape (e.g., eccentricity, solidity, etc.) and size (e.g., major and minor axis lengths, area-equivalent diameter, etc.).

For fluorescence measurements, the SwisensPoleno uses LEDs at 280 and 365 nm and a 405 nm laser diode for fluorescence excitation. The excitation sources are collimated (405nm) or focused (280nm, 365nm) and filtered using bandpass filters to narrow their emission spectrum around their center wavelength. The wavebands for detecting fluorescence emission are 333-381 nm, 411-459 nm, 465-501 nm, 539-585 nm, and 658-694 nm (referred to further using their center wavelengths: 357, 435, 483, 562, and 676 nm). Thus, the combination of excitation sources and measurement channels provides 13 viable measurements for each particle, which we will refer to using the notation of $\lambda_{ex}/\lambda_{em}$ for each excitation/emission channel. Note that the $\lambda_{ex}/\lambda_{em}$=365/357 nm and 405/357 nm channels are not included in the SwisensPoleno measurement or analysis, because the fluorescence emission detection wavelengths are longer than the excitation wavelengths. The instrument's fluorescence system covers the excitation/emission range typical for bioaerosols (Pöhlker et al., 2012). Importantly, the SwisensPoleno does not differentiate between natural particles that are inherently autofluorescent, such as some bioaerosols, and particles derived from synthetic materials such as microplastics.



The integrated instrument software makes use of a machine learning classification model for real time, single-particle classification using its holographic images. The SwisensPoleno is shipped with a default model trained by MeteoSwiss with

130 supervised learning on a subset of common central European pollen taxa and water droplets. However, users can train, evaluate, and update their instrument with a classification model prepared on other data. For this study, machine learning classification models were created, trained, and evaluated in a separate Python programming environment decoupled from the instrument. The details of the machine learning models used in this study are outlined in Sect. 2.4.

To create individual particle datasets for this study, the SwisensPoleno instrument inlet was coupled to a particle

atomizer (SwisensAtomizer)– also manufactured by Swisens AG – that entrains solid, dry test particles into the sample flow of the instrument in laboratory or test environments. The atomizer uses a small (~5 cm) acoustic speaker to apply mechanical vibrations of user-specified frequencies and amplitudes to a small volume of test particles (typically < 1.5 mL). The sample volume is physically in contact with the speaker, so that the acoustical vibrations are transferred directly to the test material inside the sample volume, sometimes inducing granular convection. Particles at the top of the volume are aerosolized because

of this vibration and a small amount of air is introduced into the sample volume to encourage the aerosolized particles to exit the sample volume and enter the sample stream of the SwisensPoleno instrument.

## 2.2 Materials and material preparation

A total of 15 particle types were analyzed using the SwisensPoleno instrument in a laboratory setting, assessing their fluorescence response and morphology through fluorescence spectroscopy and holographic imaging, respectively. An

145 overview of particles used in this study can be found in Table 1, including their class names, which are referred to throughout this work for simplicity. In addition, the total number of events (number of individual particles successfully detected with both holographic imaging and fluorescence) for each class are shown. The investigated particle types are categorized as "microplastic," "pollen," and "other." Five microplastic particle types were tested included polyamide 12 (PA), polyethylene (PE), polyethylene terephthalate (PET), poly(methyl methacrylate) (PMMA), and polypropylene (PP), which represent

common polymers used in society and frequently found as microplastics in the environment (Koelmans et al., 2022; Plastics Europe AISBL, 2022; Zhang et al., 2020). All microplastic particles were commercially purchased and tested in the dry state. To the best of the authors' knowledge, polymer samples used in this study are free from solvents, additives, or colorants. In addition to these MPs, other particle types tested included Arizona Test Dust, a volcanic ash sample from Iceland, water droplets, glass reference microspheres, and pollen samples from six different taxa. Although not all particle types in this study

are atmospherically relevant for ambient particle classification (e.g., glass microspheres), they were selected to represent a mixture of overlapping morphology, size, and/or fluorescence properties to assess the instrument's ability to differentiate between similarly featured aerosol particles.

Polyamide 12 (PA), also known as nylon 12 or PA12, was purchased in powder form (Goodfellow GmbH; Hamburg, Germany). The listed particle size range was 10-50 µm with a reported density of 1.020 g/cm$^3$. PA has many practical



applications, including product packaging, electrical insulating materials, and sports-related materials (Griehl and Ruesteivi, 1970) and is a common pollutant to the environment (Sun et al., 2019).

**Table 1: Overview of particles tested in this study and their properties.**

| Category | Particle type | Acronym/ Class name | Material supplier/source | Morphology | Material density [a] (g/cm³) | Maximum area-equiv. diameter [b] (µm) | Number of events |
|---|---|---|---|---|---|---|---|
| Microplastic | | | | | | | |
| | Polyamide (Nylon) 12 | PA | Goodfellow GmbH | Irregular | 1.02 | 27.46 ± 3.38 | 15933 |
| | Polyethylene | PE | Cospheric LLC | Spherical | 0.96 | 25.32 ± 7.73 | 12717 |
| | Polyethylene terephthalate | PET | Goodfellow GmbH | Irregular | 1.38 | 15.15 ± 3.64 | 6930 |
| | Poly(methyl methacrylate) | PMMA | Cospheric LLC | Spherical | 1.19 | 32.45 ± 4.40 | 8485 |
| | Polypropylene | PP | Sigma Aldrich | Irregular | 0.86 | 24.00 ± 9.05 | 8679 |
| Pollen | | | | | | | |
| | *Fagus sylvatica* | Beech | Thermo Fisher Scientific | Irregular | Unknown | 44.53 ± 2.53 | 6840 |
| | *Betula pendula* | Birch | Thermo Fisher Scientific | Irregular | Unknown | 21.83 ± 1.73 | 15503 |
| | *Poa pratensis* | Grass | Thermo Fisher Scientific | Irregular | Unknown | 25.85 ± 2.35 | 11521 |
| | *Corylus avellana* | Hazel | Thermo Fisher Scientific | Irregular | Unknown | 25.23 ± 1.71 | 10603 |
| | *Pinus nigra* | Pine | From source | Irregular | Unknown | 48.03 ± 2.60 | 8798 |
| | *Ambrosia artemisiifolia* | Ragweed | Thermo Fisher Scientific | Quasi-spherical | Unknown | 19. 93 ± 1.11 | 9102 |
| Other | | | | | | | |
| | Volcanic ash | Ash | From source (Eyjafjallajökull) | Irregular | 2.6[c] | 10.12 ± 2.40 | 6064 |
| | Mineral dust | Dust | Powder Technology Inc. | Irregular | 2.5-2.7 | 12.47 ± 4.18 | 9430 |
| | Soda lime glass microspheres | Glass | Thermo Fisher Scientific | Spherical | 2.5 | 30.67 ± 1.77 | 5801 |
| | Water droplets | Water | Ultrapure MilliQ water | Spherical | 1 | 12.73 ± 4.42 | 5666 |

[a] Provided by the manufacturer, unless otherwise noted.
[b] Defined as the diameter of a circle with the same area as the imaged particle, taking the maximum value from the two holographic images of each particle. Values represent the mean of each dataset ± one standard deviation.
[c] Schumann et al. (2011)

Low density polyethylene (PE) microspheres in the nominal size range of 10-106 µm were purchased from Cospheric LLC. (Santa Barbara, CA, USA). The reported density is 0.96 g/cm3. PE is used, for example, in reusable bags, rigid trays and containers, and agricultural and food packing films and made up approximately 14.4% of the 2022 global plastics production (Plastics Europe AISBL, 2022). Because of its high commercial use and potential environmental impact (Royer et al., 2018), PE remains a potentially important atmospheric microplastic to characterize.

Polyethylene terephthalate (PET) is one of the most common polymer types in use and has applications in textiles, beverage bottles, packaging materials, and other common uses (De Vos et al., 2021). While PET remains one of the most recyclable polymer materials (Plastics Europe AISBL, 2022), much of it ends up in the environment (Schmid et al., 2021). For this study, PET MPs were generated by milling larger PET granules (Goodfellow GmbH; Hamburg, Germany) using a Retsch ZM200 rotor mill. The MPs were sieved through a 50 µm stainless steel mesh, yielding the size fraction < 50 µm for the sample.

A sample of poly(methyl methacrylate) (PMMA) microspheres was purchased from Cospheric LLC. (Santa Barbara, CA, USA). According to the manufacturer, the density is 1.19 g/cm3 and more than 90% of the purchased PMMA microspheres



is reported to lie in the size range of 27 – 45 µm. PMMA, also known as acrylic, has a wide variety of practical uses (Ali et al., 2015), including the use as a transparent plastic alternative to glass (i.e., Plexiglas). PMMA can be found in environmental pollution (Brahney et al., 2020b; Thompson, 2004), reaching even the most remote regions of the world (Aves et al., 2022), but represents a polymer with low demand from plastic converters (Plastics Europe AISBL, 2022).

Polypropylene (PP) microplastics were produced by milling larger granules purchased from Sigma Aldrich (ref: 427888; isotactic, average Mw ~250,000). Briefly, the granules were melted into thin (~1 mm) cuboids at 180°C for 1 h then frozen at -70°C. The frozen cuboids were then milled in ice-cold ethanol for seven 30 sec cycles with a knife-mill (Retsch GmbH) and size fractionated using a vibratory sieve shaker (Retsch GmbH). The fraction taken from the vibratory sieve shaker was between 38-50 µm. This fraction was dried prior to use.

While the pollen taxa in this study represent a small subset of other fluorescent airborne bioaerosol (Pöhlker et al., 2013), pollen particles are included in this study to assess the ability for the instrument to distinguish aerosol particle types beyond those previously analyzed with the SwisensPoleno (Erb et al., 2023b; Sauvageat et al., 2020). The six different pollen samples tested in the SwisensPoleno were measured in a desiccated state; the bulk densities of these samples are unknown. *Betula pendula* (birch), *Fagus sylvatica* (beech), *Corylus avellana* (hazel), *Ambrosia artemislifolia* (ragweed), and *Poa pratensis* (grass) source materials were purchased from Allergon AB (Ängelholm, Sweden) and were introduced into the SwisensPoleno instrument using the SwisensAtomizer as described above. A sample of pine pollen presented in this study was sampled directly from a recent cutting of a flowering pine tree (*Pinus nigra*). The cutting with male flowers was placed within a sealed chamber that was continuously flushed with particle-free air and directly connected to the SwisensPoleno. Pollen shedding was encouraged by blowing air at the flowers using a small fan.

Arizona Test Dust (Powder Technology Inc., Arden Hills, MN, USA) was investigated with the SwisensPoleno for its response to a reference mineral dust sample. In figures, the class name for this sample is "dust". For this study, the A2 "fine" size fraction was tested in the instrument, where the manufacturer reports a nominal size range of up to 80 µm and composition of multiple mineral components dominated by silicates. Mineral dust and microplastics may share emission pathways (Brahney et al., 2021), and the use of mineral dust in this study represents a particle type with similar size and morphological features as microplastic fragments. Mineral dust particles contain a variety of mineral compositions which depend greatly on their geographical location (Engelbrecht et al., 2016), some of which have been shown to autofluoresce (Savage et al., 2017). The autofluorescence of Arizona Test Dust has previously been measured (Pöhlker et al., 2012), which showed relatively low autofluorescence intensity with no discernable spectral features.

A sample of volcanic ash was collected following the 2010 volcanic eruption of Eyjafjallajökull on Iceland. This polydisperse sample represents an additional coarse-mode particle type with similar morphology and size to microplastic fragments found in the atmosphere.

Water droplets were produced through the nebulization of Milli-Q ultrapure water using a medical nebulizer. Ultrapure water is expected to have no fluorescence and thus can serve as a baseline fluorescence measurement in the



SwisensPoleno. In addition, the spherical morphology presents an opportunity to test classification accuracy alongside other
spherical or quasi-spherical particles.

   Glass microspheres, purchased from Thermo Fisher Scientific Inc., represent a common NIST-traceable particle
standard for use in aerosol instrument calibration and testing (Pinnick et al., 1981; Dollner et al., 2023). Here, we tested glass
microspheres with a nominal mean diameter $30 \pm 1.9$ µm as reported by the manufacturer. While the fluorescence information
of glass microspheres will not be relevant for ambient coarse-mode aerosol monitoring, the microspheres share a morphology
of other common spherical microplastic beads used in, for example, personal care products (Rochman et al., 2015) and will
provide useful information in assessing the instrument's ability to discern different quasi-spherical particles.

## 2.3 Dataset creation and cleaning

The SwisensAtomizer was physically coupled to the inlet system of the SwisensPoleno, and each class of tested particles was
introduced into the instrument by adjusting the atomizer's vibrational frequency and amplitude and amount of air introduced
into the sample volume. Particles were generated in this manner for each particle type until a suitable number (> 5000) of
particles were successfully detected by both the holographic imaging and fluorescence systems, referred to henceforth as an
event. A total of 142,072 events were used in this study. After a dataset for one particle type is recorded, further processing is
needed to filter unwanted events from the dataset. These unwanted events, for example, can include: events for which the
particle lies outside a suitable position for holographic image reconstruction, which results in a blurred, out-of-focus particle
image; events clearly consisting of particle aggregation; or, unambiguous contamination by particles of types not intended to
be measured, visible through holographic imaging or detectable through unexpected fluorescence spectra of individual
particles. For example, while training for the mineral dust dataset, a pine pollen particle event can be unambiguously filtered
out due to its distinct shape and fluorescence response measured by the SwisensPoleno. During dataset preparation for machine
learning training and testing, corrections for stray light (i.e., measurements without particles present in the measurement
volume) are applied to individual events in each dataset. Further details about the SwisensPoleno fluorescence measurement
system can be found in (Graf et al., 2023).

   The distribution of events among the particle types, along with a count distribution of each particle's maximum area-
equivalent diameter (defined as the diameter of a circle with the same area as the imaged particle, taking the maximum value
from the two holographic images of each particle), is illustrated in Figure 1.






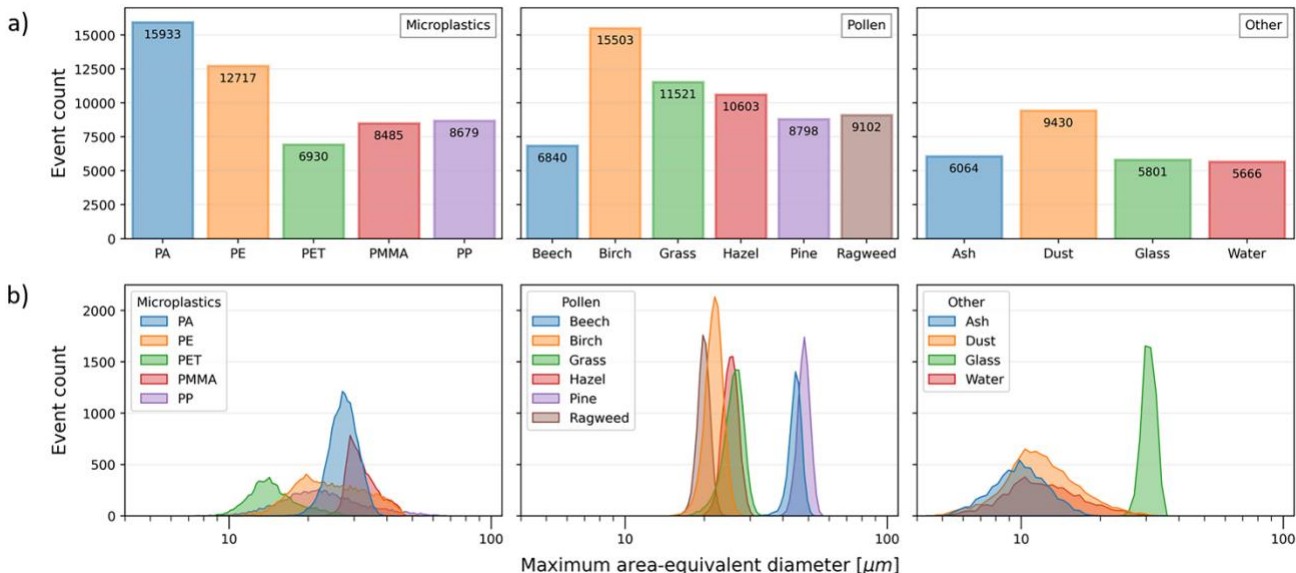

**Figure 1: Count distributions by a) class and b) size for each particle type. The maximum area-equivalent diameter is defined as the diameter of a circle with the same area as the imaged particle, and the maximum is taken from the two holographic images from each event.**

**2.4 Machine learning**

The combination of measurement methods from the SwisensPoleno creates a unique set of data for each particle event. These particle event data can then be used for training a supervised machine learning classification model to predict particle types in near real-time. A supervised machine learning classification model is one that maps predefined, discrete categories or classes to the input data corresponding to that output (Müller and Guido, 2016); in this study, the input data is represented by the two holographic images and/or the fluorescence spectra for each particle, and the output is the known particle type from that event.

The SwisensPoleno has already demonstrated high accuracy pollen taxa classification using its holographic imaging system and supervised machine learning classification model (Sauvageat et al., 2020), and, by combining holographic images with more information, such as fluorescence, the classification accuracy of pollen can become more accurate (Erb et al., 2023b). This is especially important if the features that are used to describe the particle overlap across different particle types, such as particle autofluorescence (Pöhlker et al., 2012). In such cases, the use of machine learning can be particularly useful to find

relationships between particle type and measured particle data that traditional analysis methods cannot distinguish.

In this study, two convolutional neural network (CNN) models and a multi-layer perceptron (MLP) model were trained and tested using the Keras (Chollet, 2015) and TensorFlow (Abadi et al., 2016) frameworks in the Python programming language to understand the ability of the SwisensPoleno's single particle holography and fluorescence measurements to accurately predict particle type. One CNN model ("Holo.-Only") used only the two holographic images of a particle as input,

an MLP model used only the fluorescence spectra as input ("Fl.-Only"), and the third model used both images and fluorescence as input ("Holo.+Fl."). Each of the three models were evaluated on the same set of particle events. The two CNN models that



contained the holographic images as an input layer additionally employed transfer learning using EfficientNet (Tan and Le, 2019) to improve model performance by increasing generalization and efficiency by greatly reducing resources needed for training. The dataset for this study was divided into training and testing subsets using a random 60/40% split. This partitioning

resulted in 56,824 events distributed across the 15 datasets that were subsequently used for model evaluation. Classification accuracy was evaluated using a weighted average f-score (Müller and Guido, 2016), which will be reported as an accuracy in this work. Further details of model architecture and other specifications can be found in the supplemental information.

## 3 Results

### 3.1 Morphology through digital holography

Figure 2 shows two representative events acquired using the instrument's imaging system for each particle type, displaying the range of particle sizes and morphological features used in this study. The maximum area-equivalent diameter means (± one standard deviation) for each class are shown in Table 1. The distributions of particle measurements for data of each class, including the maximum area equivalent diameters, maximum major axis lengths, maximum eccentricity, and maximum solidity, are shown in Supplemental Figures S1-S4, respectively. The Ash particle type represented, on average, the smallest

particles measured in this study with a mean maximum area-equivalent diameter of $10.12 \pm 2.40$ µm; pine particles contained the largest mean size with $48.03 \pm 2.60$ µm. However, the PP MP class had the largest single particles and greatest range to their measured size, with major axis lengths ranging from ~5-100 µm. Despite sieving during sample preparation, the milling of PP and PET particles from larger granules yielded an unexpectedly large number of particles < 10 µm, which – because the samples were untreated – may have aggregated to form large clusters to create the resulting wide size distributions. PE, PMMA,

ragweed, glass, and water particle types represent (quasi-)spherical particles tested in this study, while PA, PET, PP, ash, mineral dust, and the remaining "pollen" types are non-spherical and irregular in shape. The eccentricity (a measure of how elliptical a particle is, where a value of zero indicates a circle and values approaching 1 indicate a particle is becoming more elliptical) of PMMA, glass, PE, water, and ragweed are lowest among the different types, with mean minimum eccentricities of $0.16 \pm 0.05$, $0.16 \pm 0.06$, $0.22 \pm 0.08$, $0.25 \pm 0.12$, and $0.25 \pm 0.08$, respectively. PP, PET, mineral dust, and ash types

represent irregular, asymmetric, and rough-edged particles and their size distributions are similar to each other but much broader compared to other types (Figure 1). Solidity, a measure of a particle's 2-D projected roughness (Sinkhonde et al., 2022; Liu et al., 2015), for PP, PET, mineral dust, and ash is the lowest of all types ($0.91 \pm 0.04$, $0.91 \pm 0.04$, $0.92 \pm 0.04$, and $0.94 \pm 0.03$ respectively). As expected, the various pollen types tested were more homogenous in morphology compared to other types, indicated by their narrow maximum area-equivalent diameter size distribution (Figure 1).





**Figure 2: Representative holographic images of two particles from each particle category and each particle type. For each valid imaging event, two images are produced per particle, labeled here as "Holo. 0" and "Holo. 1". Each image is 200x200 pixels at 0.595 μm/pixel. 100 μm scale bars are shown for each image.**

### 3.2 Absolute fluorescence spectra

The mean absolute fluorescence response as measured by the SwisensPoleno for the different particle types is shown in Figure 3. Here, the water dataset is used as a proxy for the baseline fluorescence response of the instrument (Graf et al., 2023), as the ultrapure water is expected to have no detectable autofluorescence beyond an instrument background signal.

The "other" category of particles (i.e., ash, mineral dust, glass, and water), show generally low and featureless fluorescence across the excitation/emission channels. The glass microspheres have an enhanced fluorescence response in all channels with the 280 nm excitation source and in the $\lambda_{ex}/\lambda_{em}$=405/676 nm channel, which has been shown to be non-negligible in a previous investigation (Boiko et al., 2015). Mineral dust shows a slightly enhanced fluorescence response above the baseline, broadly spread across excitation and emission channels, coinciding with a previous investigation (Pöhlker et al., 2012). The ash sample displayed little to no fluorescence above the water baseline values.




**Figure 3: Mean absolute fluorescence intensity (Volts) measured by SwisensPoleno for all particle classes, where error bars are omitted for plot clarity. Columns represent three excitation sources, and the x-axis of each subplot shows center wavelengths of emission channels (not to scale). In each subplot, the "water" class represents instrument baseline fluorescence and a logarithmic y-axis used. a) "Other" category, b) "Pollen" category, and c) "Microplastic" category, where the enhanced fluorescence of MP particles in the 280/375 nm excitation/emission can be seen, several orders of magnitude above the "water" baseline signal.**



Pollen particles show an enhanced fluorescence response in all channels. For the 365 nm and 405 nm excitation sources, the average fluorescence response is more similar among the pollen types, exhibiting a broad "hump" across detection wavelengths where intensities are largely highest in the 483 nm emission detection channel. Generally, the grass pollen (*Poa pratensis*) showed the highest absolute signal response compared to other pollen species, similar to previous studies (Lichtenthaler and Schweiger, 1998; Pöhlker et al., 2013).

For MPs, the mean fluorescence in the $\lambda_{ex}/\lambda_{em}=280/357$ nm channel exhibits the highest response compared to the other particle types tested, where the absolute intensity is several orders of magnitude higher than the baseline (water) signal. Conversely, the signal from the 658–694 nm waveband for all excitation sources was about an order of magnitude lower for MPs compared to the tested pollen species. Thus, for the 280 and 365 nm excitation sources, the mean intensity of the absolute fluorescence signal decreased with increasing wavelength. For the $\lambda_{ex}/\lambda_{em}=280/357$ nm channel, the mean measured absolute
fluorescence response for polyethylene terephthalate (PET) was highest ($0.41 \pm 0.19$ V) compared to other datasets tested. For the other two excitation sources, the highest absolute fluorescence response among MPs was from the PP class. However, this is due to the largest particles found in the PP dataset, where the particle size has a direction proportionality to its fluorescence (Hill et al., 2002). In order to address this and other dependencies, the SwisensPoleno calculates a relative fluorescence for each detected particle.

**3.3 Relative fluorescence spectra**

Fluorescence measurements should ideally depend only on the composition of the detected particle. Differences, however, arise in absolute fluorescence intensity dependent on particle size, location within the measurement volume, or small differences in instrument detection capabilities and optical arrangement. These factors also make comparisons between instruments more difficult. To address this issue, the measured absolute fluorescence intensity of each channel in the
SwisensPoleno is normalized by the sum of all five detector intensities for each excitation source, providing a measured relative fluorescence value that varies from 0 to 1. A detailed explanation of fluorescence measurements with the SwisensPoleno and cross-instrument validation of the relative fluorescence spectra is described in (Graf et al., 2023).

Figure 4 details the differences between absolute and relative fluorescence for the 280 nm excitation source across all detection wavebands for the five MP particle types. The size dependence for this excitation source and measurement
channels of the absolute fluorescence shows a power law relationship with the measured intensity; that is, the relationship between absolute fluorescence intensity and size is linear in log-log space, and the slope of this relationship typically varies between ~2-3 (Hill et al., 2015; Könemann et al., 2018). This relationship holds for all MPs tested in this study except for PET, which has a slope of ~1.5 for the 280 nm excitation source response. After applying the normalization technique to calculate a relative fluorescence, the size dependence (among other non-idealities) was largely eliminated from the measurements
(Figure 4b).



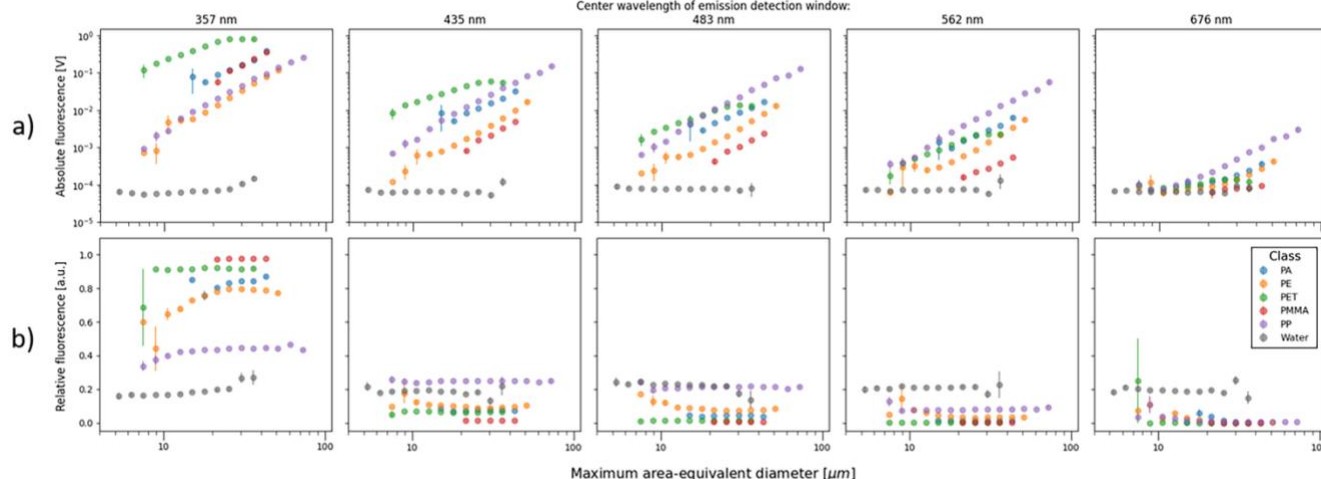

**Figure 4: a) Absolute and b) relative fluorescence of the 280 nm excitation source for MP classes and water droplets, indicating how the size dependence of the fluorescence is mostly eliminated using a relative metric. For each dataset shown, the fluorescence values are averaged for each discrete size bin, and error bars represent the calculated standard error for the means in each bin.**

The mean relative fluorescence response for the various tested particle types is shown in Figure 5. The relative fluorescence spectra for MPs exhibit a noticeably higher response in the $\lambda_{ex}/\lambda_{em}$=280/357 nm channel compared to other particles tested, which do not display this spectral feature: the mean $\lambda_{ex}/\lambda_{em}$=280/357 nm relative fluorescence values for MPs are greater than ~0.44, whereas for all other particle types tested, the mean values less than 0.33. Across all excitation/emission channels, the mean relative fluorescence values for the tested pollen types remain below ~0.5, indicating that no one channel

contributed to a majority of the spectral response of the respective excitation source. Because water, ash, dusts, and glass particles exhibit relatively low fluorescence and little variation across the detection wavelength bands, their relative fluorescence spectra are generally flat.





**Figure 5: Relative fluorescence intensities for each particle type category, where the spectral "signature" of the various particles tested is more apparent, where error bars are omitted for plot clarity. a) "Other" category, b) "Pollen" category, and c) "Microplastic" category.**





The relative fluorescence spectra represent 13 pieces of data for each valid event in the SwisensPoleno, and the ability to discern common patterns and relationships in this multidimensional dataset become difficult. We employed Uniform Manifold Approximation and Projection (UMAP) analysis (McInnes et al., 2018) to better understand the similarities and
differences of the relative fluorescence spectra. UMAP is a nonlinear dimensionality reduction technique that aims to preserve the local and global structure of high-dimensional data in a lower-dimensional space (McInnes et al., 2018). The algorithm builds a weighted nearest-neighbors graph, where the weights of the connections are determined by the local density of points and their distances in the original high-dimensional space. UMAP then optimizes the embedding by finding a low-dimensional representation that minimizes the difference between the distances of connected points in the graph and their distances in the
lower-dimensional space, capturing the inherit, underlying structure of the data, and highlighting the relationships and similarities or differences between neighboring points. This 2D representation can then be used to aid visualization and highlight these relationships between the data. Figure 6 shows the results of the UMAP algorithm applied to the relative fluorescence for all events of each data type used in this study, projected into two dimensions. The spacing of data points in the UMAP plot reflects their similarities or differences: points that are close together indicate that they are more similar based
on their spectral characteristics or fluorescence spectra; conversely, points that are far apart in the UMAP plot suggest greater dissimilarity or differences in their spectral properties. As expected, events from each dataset form relatively tight clusters, and datasets which share relative fluorescence spectral features have clusters in the UMAP that are close together or overlap. For example, water, ash, mineral dust, and glass particles overlap in the center of the plot, indicating that their relative fluorescence spectral features also overlap. The birch and hazel pollen datasets share similar relative fluorescence spectral
shapes (Figure 5b), and this is reflected in the UMAP representation with slightly overlapping clusters. For all other particle types, clustering in the UMAP plot is more distinct, which leads to the interpretation that the underlying relationships in the relative fluorescence spectral features are also quite distinct from one another.



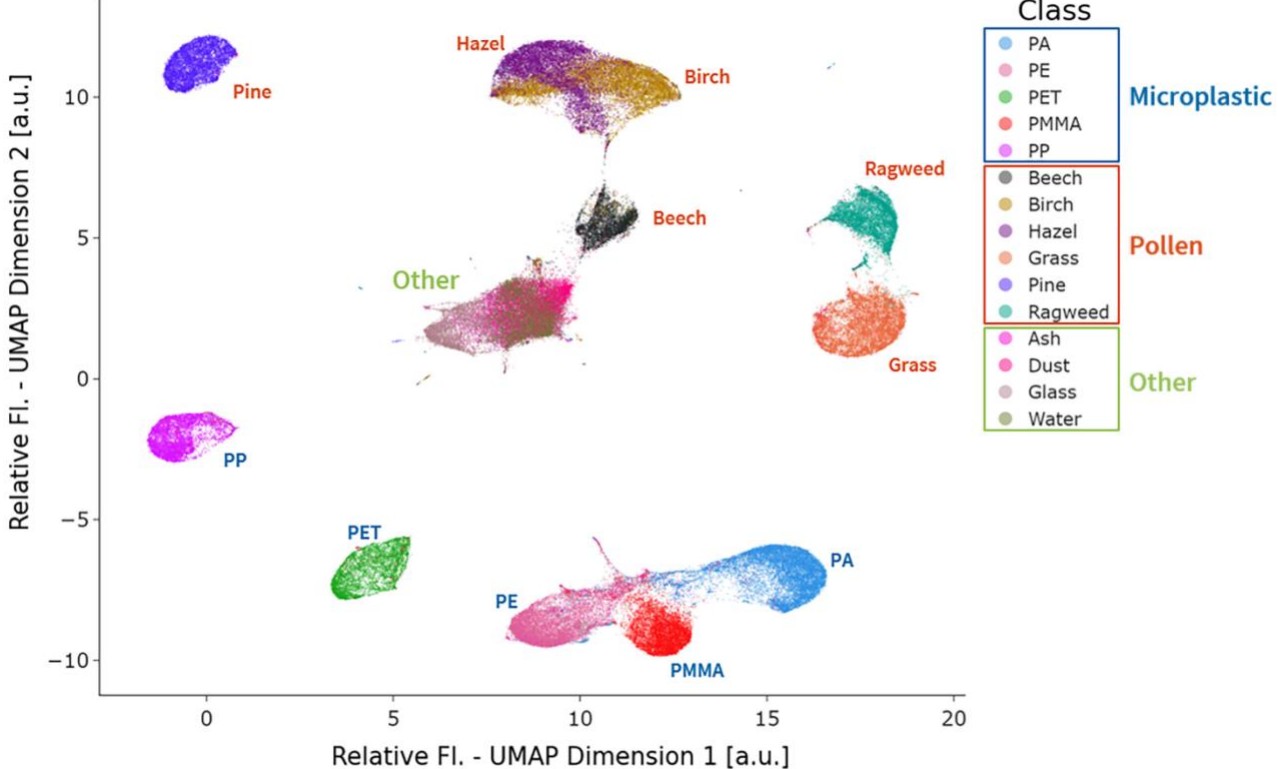

**Figure 6: UMAP plot of the relative fluorescence spectra for every event in this study. The UMAP analysis depicts the high-dimensional relative fluorescence spectra in a low dimensional (2D) representation, where each dot represents one event in the study. This 2D representation also provides insight into the relative similarity and difference between the relative fluorescence spectra: the closer each event is, the more similar their relative fluorescence spectra; conversely, and events that are further apart represent relative fluorescence spectra that are more dissimilar. Each dot is colored according to its class name in the legend; the text for each class is colored according to the category of particle types. The events from the particle classes in the "other" category (i.e., ash, dust, glass, and water) are clustered and overlapping near the center of the UMAP plot, indicating the underlying similarity of relative fluorescence in this study.**

## 3.4 Particle classification using machine learning

An integrated component of the SwisensPoleno workflow is the ability to classify particle type in near-real time by applying a trained machine learning model. This capability was assessed using the measurements in this study by employing three different machine learning model architectures utilizing holographic images and relative fluorescence spectra of the particles as input parameters for particle type classification.

The first model investigated uses a convolutional neural network (CNN) that employs only the two holographic images as input ("Holo.-Only"). This model differs from models used in previous studies of bioaerosol identification (Sauvageat et al., 2020; Erb et al., 2023a) by expanding classified particle types beyond bioaerosol and evaluating a different model architecture. Supervised learning classification models often employ the use of a confusion matrix to convey model performance. The values in a normalized confusion matrix show the classification or misclassification for different classes in





a classification model, where the values are expressed as percentages or proportions relative to the total number of particles in each true class. The diagonal values represent the correct classification for each class, while the off-diagonal values represent the misclassification percentages. The confusion matrix and performance for the Holo.-Only model can be seen in Figure 7.

The model training resulted in an overall accuracy of 90% on the test dataset. Particle types that share size and shape characteristics perform worse than those with defining features, such as pollen. For example, ash, mineral dust, hazel, PET, and PP particle types had an individual classification accuracy less than 81%, resulting from their shared irregular morphologies and/or similar size distributions. PET particles were incorrectly classified in 21% of the 2,773 events used in the test dataset as either ash or mineral dust particles, while PP was incorrectly classified as PET in 12% of the 3,458 test dataset

events. Interestingly, the spherical particle types (glass, water, ragweed, PE, and PMMA) performed surprisingly well (accuracy > 96%) considering the overlap in general morphological characteristics and size of the tested particles. Of the pollen types, hazel particles were most frequently classified incorrectly with an accuracy of 76%, where nearly all misclassified particles (23%) were classified as birch, highlighting an existing challenge in identifying these two particular pollen taxa based on their very similar morphology alone.

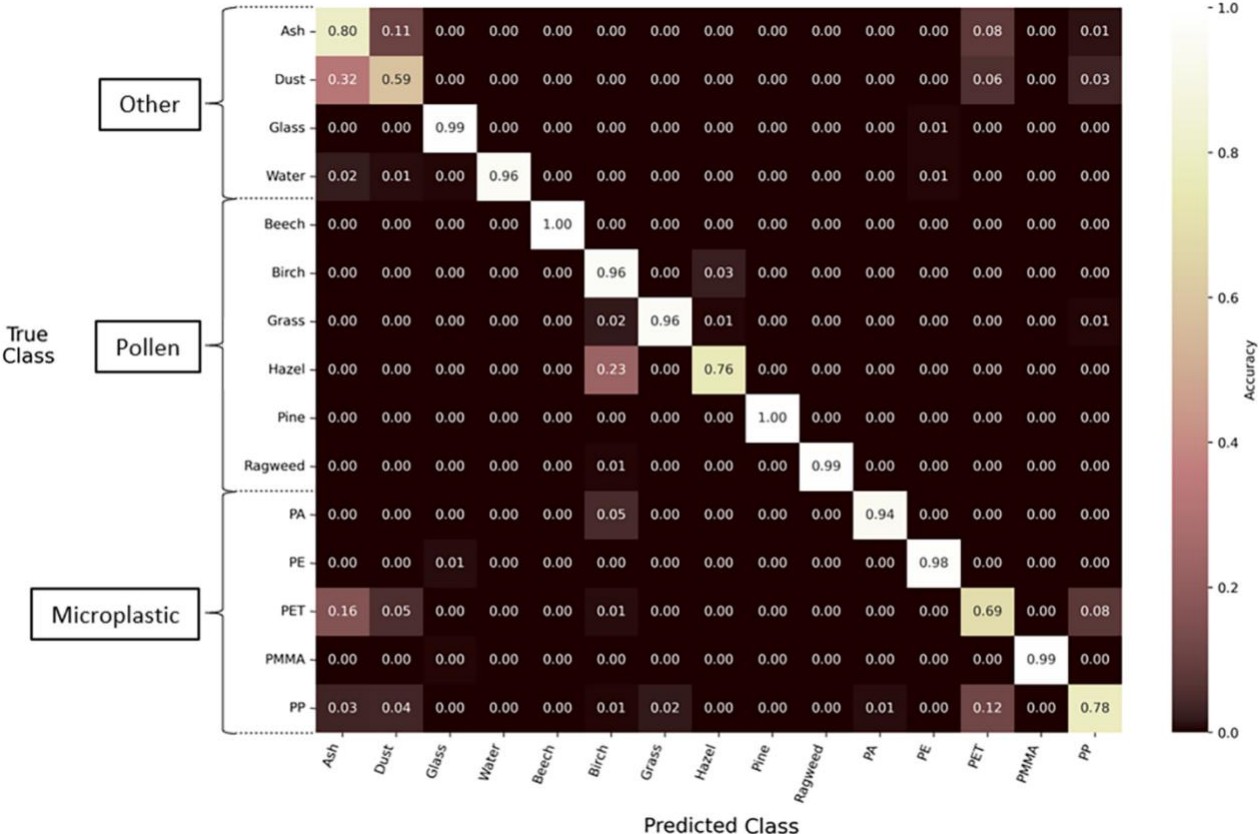


**Figure 7: Performance of the Holo.-Only machine learning model using a normalized confusion matrix. The diagonal values in the matrix represent the proportion of true positives, or the percentage of correctly classified particles for the respective true class on**



**the y-axis. The off-diagonal values represent false positives, indicating the misclassification of particles into respective predicted classes on the x-axis. The matrix is normalized along each row.**

The second machine learning classification model was a multi-layer perceptron using only the relative fluorescence spectra as input ("Fl.-Only"). Here, the Fl.-Only model had an overall classification accuracy of 94% and the distribution of prediction accuracy is shown in Figure 8. The accuracy for all pollen and MP particle types was greater than 92%, improving on deficiencies when using only the Holo-Only model for these classes. When assessing MP particle types alone, the Fl.-Only model performed with greater than 98% accuracy. In contrast, the accuracy for correct classification of water, ash, mineral

dust, and glass particles had a mixed performance, with an accuracy of greater than 95% for glass particles, but less than 74% for ash, mineral dust, and water particles.

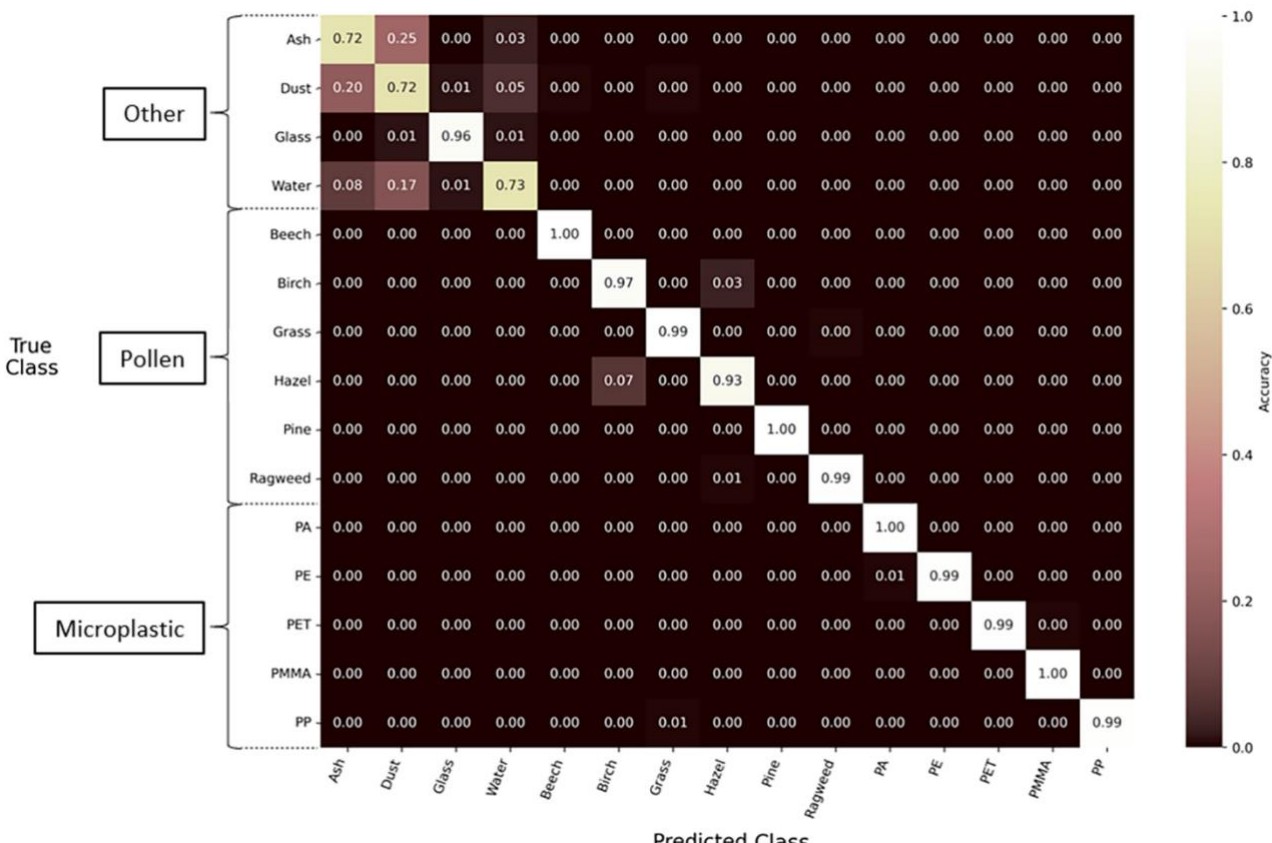

**Figure 8: Performance of the Fl.-Only machine learning model using a normalized confusion matrix. The diagonal values in the matrix represent the proportion of true positives, or the percentage of correctly classified particles for the respective true class on**
**the y-axis. The off-diagonal values represent the misclassification of particles into respective predicted classes on the x-axis. The matrix is normalized along each row.**

        The third model tested combined the holographic images and relative fluorescence approaches into a single, multi-input model ("Holo.+Fl."). An overall prediction accuracy of 98% was found for this model when using the particle types tested in this study. Figure 9 shows the normalized confusion matrix for these results, indicating the prediction accuracy across



all particle types. An accuracy of less than 95% was observed only for the ash and mineral dust particle types (85% and 82%,
respectively). All MP particles were correctly classified at least 98.5% of the time. Comparing the classification accuracies in
Figures 8 and 9, all particle types improved their classification performance compared to the models using only their relative
fluorescence or holographic images.

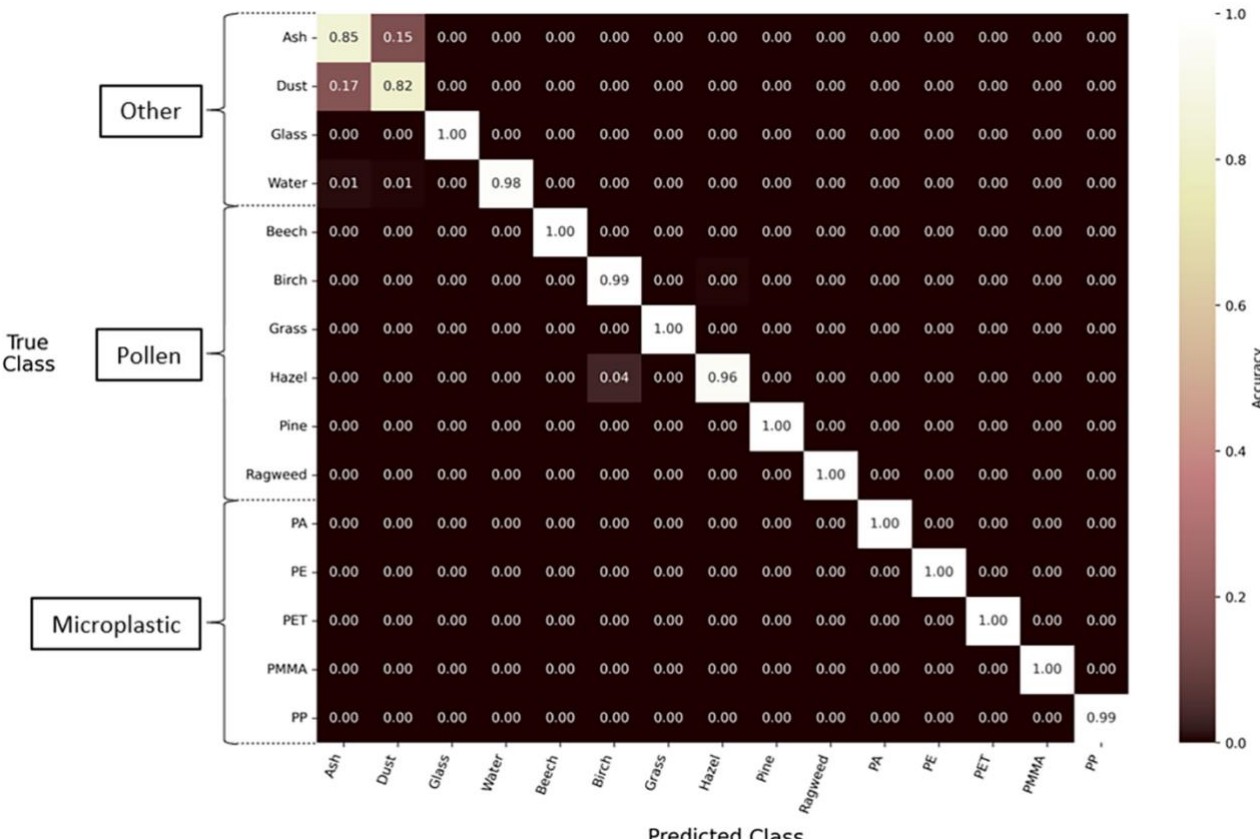

**Figure 9: Performance of the Holo.+Fl. machine learning model using a normalized confusion matrix. The diagonal values in the
matrix represent the proportion of true positives, or the percentage of correctly classified particles for the respective true class on
the y-axis. The off-diagonal values represent false positives, indicating the misclassification of particles into respective predicted
classes on the x-axis. The matrix is normalized along each row. For example, 4% of all hazel particles in the test dataset are
misclassified as birch.**

**4 Discussion**

Digital holography can provide improved information about aerosol particle size and shape beyond other light scattering
methods (Berg et al., 2017) and has been demonstrated for various coarse-mode particles, including bioaerosol (Sauvageat et
al., 2020; Erb et al., 2023a), ice crystals (Touloupas et al., 2020), and more (Berg et al., 2017). The SwisensPoleno is a powerful
instrument to capture a diverse range of single-particle morphology in near real-time. The MP particles tested in this study
closely represent two common MP morphologies – spherical beads and fragments – found in the environment (Cowger et al.,



2020; Helm, 2017; Yu et al., 2023). However, particles that share morphological features and size distributions may be misclassified by a machine learning model that uses 2-D images as the only training data input, as was demonstrated in this work. For example, fragmented, irregular particle types in this study that had similar size distributions – such as PP, PET, volcanic ash, and mineral dust – performed with lower accuracies (accuracies < 81%) when using a machine learning model employing holographic images as the only input, where including additional, concurrent measurement information may increase accuracy of real time particle identification. On the other hand, spherical and quasi-spherical particle types – such as ragweed pollen, water droplets, glass beads, PE microspheres, and PMMA microspheres – performed well (accuracies > 96%) when considering their holographic images only, indicating that this machine learning model can find distinctive features not easily identifiable by eye.

Polymers have been known to have autofluorescent properties, but these properties may complicate analyses using other instruments or techniques (Asfour et al., 2020) and thus may not be considered as identifying features. Although the characterization of MP autofluorescence has recently been shown to be a promising tool for identifying polymer type (Monteleone et al., 2021a, 2021b), until now, no instrument has shown the ability for near real-time identification of airborne MPs. The MP particles tested in this study have an absolute fluorescence response greater than or on the same order as pollen particles. The exceptionally strong fluorescence observed for PET particles aligns with expectations, as PET contains an aromatic ring in its composition acting as a strongly emitting fluorophore. PET MPs and nanoplastics have previously been observed to exhibit autofluorescence, due to their strong absorption in the UV region (Lionetto et al., 2022). And, while PET exhibits fluorescence when excited at longer wavelengths (i.e., in the visible spectrum), the results from this study showed that as the excitation wavelength increases, the fluorescence intensity decreases. Additionally, On the other hand, polymers which lack aromatic or highly conjugated double bond structures (i.e., PA, PMMA, PP, and PE) are not traditionally associated with strong autofluorescence (Shadpour et al., 2006); nonetheless, PA, PMMA, PP, and PE microplastics used in this study displayed fluorescence intensities on the same order as the primary biological particles tested. These results may suggest the presence of other factors that contribute to their measured fluorescence, such as the unintended presence of impurities or additives (i.e., non-intentionally added substances; (Bridson et al., 2023)), and further investigation is required to understand the specific mechanisms driving the fluorescent properties observed.

While the autofluorescence properties of other airborne particles (such as polycyclic aromatic hydrocarbons, mineral dust, pollen, etc.) may overlap (Pöhlker et al., 2012; Savage et al., 2017), the use of the SwisensPoleno instrument is a very promising method to overcome the challenge of distinguishing MPs from other airborne particles due to the combined information of particle morphology and fluorescence provided by the instrument. The relative fluorescence spectra for the tested particles show distinct spectral features that can be distinguishable from each other, as demonstrated with, for example, the UMAP dimensionality reduction technique (Figure 6). The relative fluorescence measurement system, combined with a machine learning classification model, allows for particles that share morphological characteristics to be distinguished with a high degree of accuracy, such as the spherical particles used in this study (water droplets, ragweed pollen, and glass, PE, and PMMA microspheres). When using the relative fluorescence of the particles in a machine learning model, the overall



classification accuracy was enhanced compared to when particle holographic images were only used for model inference, increasing from 90% to 94%. Particles that exhibit a distinct fluorescence spectral pattern can be differentiated from other particle types with high accuracy using the machine learning model; conversely, particles that have low relative fluorescence and non-distinct spectral features – such as water and mineral dust – were more often misclassified in model evaluation. This result could prove problematic for any ambient measurements that rely strictly on fluorescence in environments where the

interaction of water droplets and mineral dust are possible. Here, too, future work using the SwisensPoleno may help classify these ambiguous fluorescence events by including polarized scattering information for each event.

It is important to acknowledge that the atmosphere contains a wide variety of aerosols in terms of composition, size, and shape (Seinfeld and Pandis, 2016). This study only considers specific subsets of particle types that the SwisensPoleno instrument might encounter during ambient monitoring; therefore, while the machine learning models in this study exhibited

generally high classification accuracy, generalizing them to ambient measurements with the SwisensPoleno likely will lead to misclassifications. For example, spores of various bacteria and fungi – known to be an important atmospheric bioaerosol that autofluoresce (Hill et al., 2009) – are not considered here and would certainly be misclassified if the models used in this study – lacking the necessary training data – were used in ambient particle identification. For MPs, while the MPs tested in this study were assumed to be without additives, many plastics are produced with additives that enhance their performance or

functionality (Hahladakis et al., 2018). Thus, it can be assumed that much of the MPs in the environment also contain additives, which could alter their measured fluorescence in the SwisensPoleno. Thus, further investigation is required to understand how components of airborne microplastics found in the environment – such particles comprised of multiple components (i.e., tire and road wear particles; (Kreider et al., 2010)), those containing pollutants adsorbed onto the surface (e.g., (Fu et al., 2021); (Gao et al., 2021)), or those that have undergone environmental weathering processes such as photooxidation (Sun et al., 2020)

– contribute to changes in measured fluorescence and how this may impact their measurement in the SwisensPoleno.

## 5 Conclusions

In this study, the high-performance capability of the SwisensPoleno was demonstrated to characterize and classify MP particles and compare the classification performance against similarly featured coarse-mode aerosol particles, including mineral dust, various pollen taxa, and water droplets. Under laboratory conditions, the instrument characterized and identified – with high

classification accuracy – MP particles in near real time using holographic images and fluorescence spectral analysis for single coarse-mode particles.

The microplastics tested in this study represent common polymer types for microplastics found in environmental pollution. They display sufficient fluorescence intensities that can be measured with the SwisensPoleno, and have distinct spectral features, aiding in distinguishing particle type among both MPs and non-MPs. In the machine learning classification

model configurations used in this study, model performance increased when combining holographic images of single microplastic particles with their measure relative fluorescence, expanding on previous studies using the instrument for

bioaerosol identification. Future work is required to understand how increasing sample complexity can affect instrument performance and particle typing accuracy. For example, more particle types with varying morphologies and compositions need to be tested, such as MP fibers, MP particles that have experienced atmospheric processing or weathering, and MP particles

with additives or other chemical composition differences. The prediction accuracy of these various other MPs needs to be evaluated alongside other autofluorescing aerosol particles, including further bioaerosol types such as spores, PAHs, combustion byproducts, and tire and road wear particles.

While an improvement to the comprehensiveness of the data used can improve future studies, all MPs tested in this study demonstrated detectable fluorescence, falling within the measurement range of the SwisensPoleno. The combination of

fluorescence and holographic imaging enabled the machine learning models to distinguish various MP types from one another and other coarse-mode particles in the study, suggesting the potential suitability of the instrument for monitoring airborne MPs in ambient conditions. The ability to monitor and accurately classify MPs in situ and in near-real time would provide a substantial increase in understanding of the abundance, distribution, properties, and potential impact MP particles could have on humans and the environment.

**Author contribution**

NDB and BW conceptualized the study. NDB, JB, and BW designed the experiments. NDB, JB, and LAD prepared materials for measurement. NDB and JB performed the measurements. EG and YZ developed the data processing and machine learning software, which was edited and implemented by NDB. NDB analyzed the data and prepared the manuscript with contributions from BW and LAD, and all co-authors contributed to the reviewing and editing of the manuscript.

**Competing interests**

EG and YZ are employees of Swisens AG, yet their employment at Swisens AG does not impact the results presented or influence the interpretation of the results. The research was conducted with good scientific rigor and impartiality. The authors have no other competing interests to declare.

**Acknowledgements**

The authors acknowledge the support of Hans Moosmüller for supplying the Arizona Test Dust sample and Erny Niederberger and Maximilian Dollner for their valuable discussions regarding study design and manuscript preparation. Additionally, the authors acknowledge and thank Itziar Otazo Aseguinolaza and Lukas Wimmer for their assistance in preparation of the polypropylene and polyethylene terephthalate samples used in this study. The authors acknowledge funding through the University of Vienna and the Gottfried-and-Vera Weiss Foundation and FWF in the context of the PlasticSphere project





(AP3641721). The SwisensPoleno instrument was purchased under the University of Vienna's investment program (IP734015). LAD would like to acknowledge funding from the IMPTOX project (European Union Horizon 2020 program grant number 965173).

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
