# Peer review of "Merging holography, fluorescence, and machine learning for in situ, continuous characterization and classification of airborne microplastics"

_EGUsphere, 2023_

## Author Response (AR1)

**Reply to the review by Anonymous Referee #1 for the manuscript, "Merging holography, fluorescence, and machine learning for in situ, continuous characterization and classification of airborne microplastics" by N. D. Beres et al.**
* * *
We thank the anonymous reviewer for their thorough and thoughtful responses and recommendations to improve the manuscript. Below, questions and comments by the reviewer are in blue and the responses by the manuscript authors are in black. Note that line numbers referred to here in the authors' responses correspond to those in the revised manuscript unless otherwise stated.
* * *
Page 3, line 75 on statement "Some commercially available polymers have previously been examined for their autofluorescence": It might be helpful for the readership to add a few sentences already in the introduction on the molecular motifs that are responsible for the fluorescent emission. Especially for the non-aromatic polymers, which one typically would not consider efficient fluorophores, the observed strong emission reported already in previous studies is an interesting phenomenon.

The authors agree that this could be helpful for readers. We've modified the introduction (Section 1, Lines 74-80) to include some information about the reasons for fluorescence in polymers:

"An often-overlooked material property of airborne microplastics that has the potential to specify particle type is their natural ability to fluoresce, or autofluoresce, which results from the spontaneous emission of light at one wavelength by the fluorophores (molecule or compound capable of fluorescence) of the polymers from excited electromagnetic states when exposed to higher-energy, lower-wavelength light (Lakowicz, 2006). For polymers, this can be due strictly from their molecular structure containing contain aromatic rings, conjugated double bonds, or other fluorophores, from stabilizers, additives, or impurities unintentionally added to the substance during the polymerization process or after production, or by some combination thereof."

We've also included some context to fluorescence by non-aromatic polymers by modifying the discussion (Section 4, Lines 471-480):

"...polymers which lack aromatic or highly conjugated double bond structures (i.e., PA, PMMA, PP, and PE) are not traditionally associated with strong autofluorescence (Shadpour et al., 2006); nonetheless, PA, PMMA, PP, and PE microplastics used in this study displayed fluorescence intensities on the same order as the primary biological particles tested. These results may suggest the presence of other factors that contribute to their measured fluorescence, such as the unintended presence of impurities or additives (i.e., non-intentionally added substances; (Bridson et al., 2023)). Additionally, while polyolefins like

polyethylene (PE) and polypropylene (PP) do not contain fluorophores in their chemical structure, photo- or thermal-oxidation (Allen et al., 1977; Zhao et al., 2022), impurities (Bridson et al., 2023; Laatsch et al., 2023), fibers structural defects (Poszwa et al., 2016), or formation of high molecular weight (HMW) clusters (Laatsch et al., 2023) can cause PE and PP to become fluorescent. For example, during the photo-oxidation process, enones and dienones can be formed (Allen et al., 1977), which makes those polymers gain fluorescent properties."

Page 3, line 82 on statement "No study has used the intrinsic fluorescence of polymers for airborne particle identification and characterization in situ." I think the preprint by Gratzl et al., 10.26434/chemrxiv-2023-qzhr8 could/should be cited in this context.

The authors thank the reviewer for suggesting this preprint. We have removed the statement ("No study has used the intrinsic fluorescence…") and added the following to the Introduction (Section 1, Lines 92-99) to acknowledge the work by Gratzl et al.:

"One recent work has shown the promising ability to classify airborne MPs using their autofluorescence (Gratzl et al., 2024). Here, Gratzl et al. (2024) leverage the Wideband Integrated Bioaerosol Sensor (WIBS; Droplet Measurement Technologies, Longmont, CO, USA) to detect microplastics based on specific fluorescence signatures excited at two wavelengths and detected in two emission wavelength bands. While their approach provides a promising step towards a greater understanding of MPs in the atmosphere, the recent introduction of the SwisensPoleno air-flow cytometer (Swisens AG; Emmen, Switzerland), was recently shown to classify biological aerosol particles with high accuracy (Erb et al., 2023, 2024; Sauvageat et al., 2020), expanding the spectral capabilities of the WIBS, and combines additional particle information to strengthen the classification ability of MPs and other atmospheric coarse-mode aerosol."

Page 3, line 90 on statement " … assess the fluorescence response of various common microplastics": I wonder why polystyrol was not included here. It is widely used probably also shows a characteristic fluorescence due to the aromatic structure. Along these lines and in more general terms, according to which criteria were the five polymers selected.

Polystyrol (or "polystyrene") was unfortunately unavailable to the authors during data collection, and it was chosen to work with the five polymers we had available to us at the time rather than increase the complexity of this feasibility study. We agree with the reviewer that polystyrene is likely to have a strong fluorescence signal due to its aromatic nature, prevalence as an environmental pollutant, and is an important component to include in future studies.

The five polymers tested in our study were chosen based on a combination of commercial availability and those which are likely to be found in the environment. We agree that more testing is needed, expanding the breadth of polymer types as well as non-polymer types. We have addressed this need for more testing in the discussion (Section 4).

The study by Ornik et al. should be cited (and probably also discussed) somewhere https://doi.org/10.1007/s00340-019-7360-3

We thank the reviewer for suggesting this study be included in our manuscript. We have included the following information in our manuscript's introduction (Section 1, Lines 87-91):

"…Ornik et al., (2020) examined the fluorescence spectra of eight, large commercially obtained polymer samples – including polypropylene, polyethylene, polyethylene terephthalate, and two polyamides – and demonstrated that their emission spectra is generally distinguishable from non-polymer samples. They acknowledged that these same principles can be applied to microplastics of various sizes and shapes, while leveraging advanced analysis methods such as machine learning, for high accuracy classification."

Table 1: Did the authors receive any information on the age of the pollen samples? Such commercially available biological reference substances are not necessarily freshly collected, which brings up the question on how atmospherically representative the derived fluorescence signals are.

The authors acknowledge the spectrum of atmospheric relevancy of particles used in this study (Section 2.2, Lines 172-175; Section 4). The primary focus of the current manuscript is to gauge the SwisensPoleno's response to various microplastics as a promising first step towards identifying and characterizing these particles in near real-time.

While commercially available biological reference substances, like pollen, are widely used in laboratory settings, factors such as age and storage conditions (e.g., humidity) may influence their fluorescence properties. For example, Pöhlker et al. (2013) state (Page 3373), "…the fluorescence properties of commercially obtained and freshly harvested pollen samples are overall similar, except for increasing intensity with age, and that all samples are generally comparable." Understanding the effect of the age of the pollen samples used in the present study is outside the scope of this work; rather, the usage of pollen was to (Lines 209-210), "assess the ability for the instrument to distinguish aerosol particle types beyond those previously analyzed with the SwisensPoleno."

However, while not addressed in this study, the authors agree that future work should assess how the SwisensPoleno's fluorescence response, specifically, is affected by different variables, such as the source and age of various pollen taxa, including both commercially available reference pollens and freshly collected samples. We have added an additional statement in the discussion that includes the recommendation that future studies should consider age of pollen as a variable in fluorescence response in the SwisensPoleno (Lines 502-504): "…while not addressed in this study, future work should assess how the SwisensPoleno's fluorescence response is affected by different variables, such as the source and age of various pollen taxa, of both commercially available reference pollens and freshly collected samples".

Page 8, line 235 on statement "Further details about the SwisensPoleno fluorescence measurement system can be found in (Graf et al., 2023).": It is a pity that the Graf et al. reference is not available yet. It is cited few times and could be quite useful for a better understanding of the fluorescence response of the instrument. If the publication of this study still needs some time, it might be worth to put some relevant information still in this study to provide the reader a more comprehensive understanding of the technique.

The authors agree that, without the Graf et al. reference, a more comprehensive explanation of the fluorescence technique used in the SwisensPoleno instrument is needed. We will add a section to the Supplemental Information that provides an overview of the fluorescence system within the SwisensPoleno instrument. In addition, we have changed the statement to account for this (Lines 142-143): "Further details about the SwisensPoleno fluorescence measurement system can be found in the supplemental information."

Page 11, line 296 on statement "Here, the water dataset is used as a proxy for the baseline fluorescence response of the instrument": Is this the standard procedure to determine the background or is there a force trigger function as commonly used for the WIBS?

The SwisensPoleno data processing algorithm uses an ultrapure water dataset as a thresholding dataset; it does not use a force trigger function as is commonly used for the Wideband Integrated Bioaerosol Sensor (WIBS). This water dataset is then used to threshold the baseline fluorescence response during the routine processing of the measured data, because, as was stated (Line 297 in the unrevised manuscript), "…ultrapure water is expected to have no detectible autofluorescence beyond instrument background signal". As stated in the previous reviewer response, further explanation will be included regarding the fluorescence system of the SwisensPoleno in the Supplemental Information, which will also elucidate this point. The statement "Here, the water dataset is used as a proxy for the baseline fluorescence response of the instrument…" will be adjusted for clarity (Lines 314-315): "Here, the water dataset is shown to represent the baseline fluorescence response of the instrument, as the ultrapure water is expected to have no detectable autofluorescence beyond an instrument background signal."

Page 14, line 345 on statement "The relative fluorescence spectra for MPs exhibit a noticeably higher response in the λex/λem=280/357 nm channel compared to other particles tested": What is the molecular explanation for this spectral feature?

As outlined in the response to the reviewer's first comment, fluorescence emission spectral features in the examined MPs, in general, are due to their chemical structure, which may be efficient at absorbing UV wavelength light (Lionetto et al., 2022), such as aromatic functional groups or other conjugated bond systems. The presence of impurities, additives, or other substances within plastic may also alter the fluorescence response (Bridson et al., 2023; Laatsch et al., 2023). External factors, such as photo- or thermal-oxidation may alter the polymer's chemical structure, leading to the introduction of oxygen-containing groups (i.e.,

carbonyls) which may change their fluorescence characteristics. This is important, because photooxidation may occur to airborne microplastics through atmospheric aging (Ouyang et al., 2021), for example. Monteleone et al. (2021) showed that thermal treatment also alters the fluorescence characteristics of various polymers, which may modify detection abilities or spectral response in fluorescence spectroscopy applications, such as the SwisensPoleno. While future work can focus on understanding the specific excitation and emission wavelength bands that can be targeted towards environmental MP detection, delving too deep into the molecular explanation of this response is beyond the scope of this feasibility study.

References:

Allen, N. S., Homer, J., and McKellar, J. F.: Origin and role of the luminescent species in the photo-oxidation of commercial polypropylene, J Appl Polym Sci, 21, 2261–2267, https://doi.org/10.1002/app.1977.070210823, 1977.

Bridson, J. H., Abbel, R., Smith, D. A., Northcott, G. L., and Gaw, S.: Release of additives and non-intentionally added substances from microplastics under environmentally relevant conditions, Environmental Advances, 12, 100359, https://doi.org/10.1016/j.envadv.2023.100359, 2023.

Gratzl, J., Seifried, T. M., Stolzenburg, D., and Grothe, H.: A fluorescence approach for an online measurement technique of atmospheric microplastics, Environmental Science: Atmospheres, https://doi.org/10.1039/D4EA00010B, 2024.

Laatsch, B. F., Brandt, M., Finke, B., Fossum, C. J., Wackett, M. J., Lowater, H. R., Narkiewicz-Jodko, A., Le, C. N., Yang, T., Glogowski, E. M., Bailey-Hartsel, S. C., Bhattacharyya, S., and Hati, S.: Polyethylene Glycol 20k. Does It Fluoresce?, ACS Omega, 8, 14208–14218, https://doi.org/10.1021/acsomega.3c01124, 2023.

Lakowicz, J. R.: Principles of Fluorescence Spectroscopy, 3rd ed., edited by: Lakowicz, J. R., Springer US, 954 pp., https://doi.org/10.1007/978-0-387-46312-4, 2006.

Lionetto, F., Lionetto, M. G., Mele, C., Corcione, C. E., Bagheri, S., Udayan, G., and Maffezzoli, A.: Autofluorescence of Model Polyethylene Terephthalate Nanoplastics for Cell Interaction Studies, Nanomaterials, 12, https://doi.org/10.3390/nano12091560, 2022.

Monteleone, A., Brandau, L., Schary, W., and Wenzel, F.: Using autofluorescence for microplastic detection – Heat treatment increases the autofluorescence of microplastics, Clin Hemorheol Microcirc, 76, 473–493, https://doi.org/10.3233/CH-209223, 2021.

Ornik, J., Sommer, S., Gies, S., Weber, M., Lott, C., Balzer, J. C., and Koch, M.: Could photoluminescence spectroscopy be an alternative technique for the detection of microplastics? First experiments using a 405 nm laser for excitation, Applied Physics B, 126, 15, https://doi.org/10.1007/s00340-019-7360-3, 2020.

Ouyang, Z., Zhang, Z., Jing, Y., Bai, L., Zhao, M., Hao, X., Li, X., and Guo, X.: The photo-aging of polyvinyl chloride microplastics under different UV irradiations, Gondwana Research, https://doi.org/10.1016/j.gr.2021.07.010, 2021.

Pöhlker, C., Huffman, J. A., Förster, J.-D., and Pöschl, U.: Autofluorescence of atmospheric bioaerosols: spectral fingerprints and taxonomic trends of pollen, Atmos Meas Tech, 6, 3369–3392, https://doi.org/10.5194/amt-6-3369-2013, 2013.

Poszwa, P., Kędzierski, K., Barszcz, B., and Nowicka, A. B.: Fluorescence confocal microscopy as effective testing method of polypropylene fibers and single polymer composites, Polym Test, 53, 174–179, https://doi.org/10.1016/j.polymertesting.2016.05.025, 2016.

Shadpour, H., Musyimi, H., Chen, J., and Soper, S. A.: Physiochemical properties of various polymer substrates and their effects on microchip electrophoresis performance, J Chromatogr A, 1111, 238–251, https://doi.org/10.1016/j.chroma.2005.08.083, 2006.

Zhao, Y., Long, J., Zhuang, P., Ji, Y., He, C., and Wang, H.: Transforming polyethylene and polypropylene into nontraditional fluorescent polymers by thermal oxidation, J Mater Chem C Mater, 10, 1010–1016, https://doi.org/10.1039/D1TC05520H, 2022.

**Reply to the review by Anonymous Referee #2 for the manuscript, "Merging holography, fluorescence, and machine learning for in situ, continuous characterization and classification of airborne microplastics" by N. D. Beres et al.**

We thank the anonymous reviewer for their thorough and thoughtful response and their recommendations to improve the manuscript. Below, questions and comments by the reviewer are in blue and the responses by the manuscript authors are in black. Note that line numbers referred to here in the authors' responses correspond to those in the revised manuscript unless otherwise stated.

Overall I am very pleased to have reviewed the manuscript, however, I am not clear if the model trained can be used for live measurements and why there were no measurements using a combination of particles to test the system.

First: yes, models that are trained offline can indeed be used for "live" measurements when the instrument is deployed for ambient monitoring. As stated in the (unrevised) manuscript (Section 2.1, Lines 128-133), the instrument can be updated to use a model defined by end users. We have clarified the wording to be clearer: "The integrated instrument software makes use of a machine learning classification model for real time, single-particle classification using its holographic images. The model used for real-time or "live" particle classification during instrument deployment is developed, trained, and tested offline on particle types the user expects the instrument to encounter."

Second: "using a combination of particles to test the system" is not the intention of our study or particularly useful for supervised machine learning model evaluation, as it eliminates the ability to accurately assess the model's predictive power. This is because there would be no ability to know which particle type (i.e., the ground truth) is being introduced into the instrument ahead of time from this mixed sample. Instead, validating the model's performance requires a dataset where the true particle type is known beforehand, allowing for a direct comparison between the models' predictions and the actual particle types. Supervised learning models – like those used in this study – already accomplish something very similar to this during the training and testing phases for these types of machine learning models, using a "unseen" mixture of the measured data (where particle type is known ahead of time) that is set aside before training (Müller and Guido, 2016). After the model is trained, the machine learning model code uses this portion of previously set aside measured data, randomizes the order of the data, and the model makes predictions on this unseen, random-order data, and compares the predicted particle type to the actual particle type (again, known ahead of time) to get a measure of how well the model can predict on new, unseen data. The results of this comparison are displayed in the confusion matrices in our study (Figures 7, 8, and 9).

Expanding on this if the study had an example dataset to finish that had resuspended already collected environmental MPs (or mixture of the 15 tested particles) this would provide a more convincing argument for its use and is shown in examples of other novel sampling or analysis methods. It would further reinforce the later results discussion and conclusions on the SwisensPoleno system and CNN use and impact of environmental samples and their varying fl response which would affect classification.

We appreciate the possibility of resuspending a mixture of the 15 particle types tested in this study; however, we believe this would provide very little to no novel information beyond what was already shown in the manuscript for the reasons described in the previous comment response above.

Given the manuscripts main development are the models and datasets created it would be good to see this hosted in an opensource platform, referenced in the text, so that other researchers could access it and build on it.

For the machine learning models developed in this study, the model architectures were described in detail in the Supplemental Information (Section S2). The authors cannot share specific code without authorization from the manufacturer. More importantly, the measurement data, and the trained models created from them, do not represent ambient particles and any users of this information may be misled by applying them to ambient measurements. Section 2.2 outlines the particles used and discusses their atmospheric relevance, which may be limited; for example, the models include a dataset of glass microspheres, which one will never encounter in ambient measurements. We have added a statement on Data Availability (Lines 541-542): "The data and code used in this study are available for research purposes on request from the authors."

To fix/address:

Lines 3 – 33: Abstract, please include the models you used, ML is fairly ambiguous, even if you just state the most optimal model that is fine.

The following statement was added to the abstract (Lines 32-34): "The latter model, using both the holographic images and fluorescence information for each particle, was the most optimal model used, providing the highest classification accuracy compared to employing models with only the holography or fluorescence response separately."

Lines 30-33: Abstract, stating that the holo+fl model was the most optimal, this is a key result and adaption of the existing MeteoSwiss model, but is not mentioned.

We thank the reviewer for this suggestion; it was addressed in the previous comment.

Line 46: MPs are in the majority of publications classified into the 1-5,000um scale, to fit your classification us more I would also add in the newer ISO ruling on size as well as the

publication you have referenced. This is a better justification in my opinion (https://www.iso.org/files/live/sites/isoorg/files/store/en/PUB100472.pdf)

Our goal was to align with size suggestions outlined in Hartmann et al. (2019), which aims to produce less ambiguous size classes for plastic pollution; indeed, not only is Hartmann et al. (2019) cited in the ISO report, but the report's recommendation on language and size classes for microplastics coincide with those used in Hartmann et al. (2019) and, thus in our study. We thank the reviewer for providing an additional reference that strengthens this definition; we have added it to our definition of microplastics in the manuscript (Line 52).

Line 55-65: All references are fine, but none are particularly recent, there are many reviews and also studies that show the detection of MP and NP that could be used. Just an observation that doesn't have to be addressed but was surprising to see.

We thank the reviewer for this observation. We have included new articles based on both reviewers. Currently, more than half of the of the references cited in this manuscript were published in or after 2020.

Line 117-127: Fl measurement regions, further explanation of why these regions would be appreciated (not only that this is what is available through the SwissPoleno), in MP research there is some debate on what wavelength to use and filter at. This could be an extra comment added to the SI given to support researchers more deeply involved with this aspect.

The excitation wavelengths and emission detection wavelength bands for the SwisensPoleno were likely chosen to provide insight on the fluorescence information of many particle types. The spectral regions of importance for fluorescence spectroscopy and particle detection is an ongoing research topic (e.g., Pöhlker et al., 2012, 2013; Savage et al., 2017), and will likely vary based on many variables. Understanding which variables and to what degree they influence the classification accuracy is a complicated problem that is not addressed in this study, as it is outside the scope of this initial test of feasibility for the detection and characterization of pure, commercially purchased microplastic particles.

Line 128-133: Clarity on why MeteoSwiss not used, also can a custom python script be loaded to the system? MeteoSwiss can be trained and updated but is its current supervised learning method not useful for MP analysis?

As stated in the unrevised manuscript (Section 2.1, Lines 129-130), the MeteoSwiss model was trained to identify a subset of pollen taxa common to central Europe. Thus, this model would not be appropriate to identify airborne microplastic particles. To the best of the authors' knowledge, no model used on the SwisensPoleno system had been developed to identify airborne plastic particles.

Second: Yes, a custom machine learning model – that is, trained specifically to identify the particle types expected to be encountered in the atmosphere for the instrument's location – can be used. These models can be developed, trained, and tested using the Python programming language, like it was in our study.

Line 143: Materials section: I would like to confirm why PS was not considered for analysis as it is one of the major MP polymers detected in aerosols, it is also available from Cospheric and Goodfellows. A comment later in the discussion could cover these questions as other MP researchers would have. Is the statement in line 459 related to this?

The authors assume "PS" refers to the polymer polystyrene, and we agree that including polystyrene microplastic particles to test in this study could have been beneficial. Unfortunately, at the time of testing, polystyrene particles were not available to the authors. While Line 459 of the manuscript does not relate to this fact, the final paragraph of the discussion (Section 4, Lines 498-514) does address future studies and the need to include additional particle types, shapes, and sizes.

We have added "18.2 MΩ-cm" to the description of the ultrapure water droplets used in this study (Line 230). We thank the reviewer for suggesting this clarification.

Line 223: Added details on the flow rate limits and particle size limits of the system would be good to add here, or in section 2.1. The questions I am thinking while read are, is there a reason no PM2.5 particle sizes were assessed (PS would be the only standard easily availably in this size range), is it because the SwisensPoleno has a detection limit. Secondly is there a flow rate limit as well. A web search of the system does not give much results unless in it's the Mars of Jupiter variant.

The authors thank the reviewer for this suggestion. The instrument is, in fact, a Jupiter variant of the SwisensPoleno system, as was stated in the abstract (Line 22) as well as in the introduction (Line 104) of the unrevised manuscript. However, we have also added this into the description of the instrument (Section 2.1, Line 111) of the revised manuscript. In addition, we have added the stated size limits and effective flow rate of the instrument in this section as well (Lines 118-120), according to the manufacturer specifications: "According to the manufacturer, the instrument has an effective flow rate of 40 LPM and particles can be detected in their multi-sensor system between that are in the size range 0.5-300 μm.."

Aerosol particles less than or equal to an aerodynamic diameter of 2.5 μm ("PM2.5") were not intentionally omitted from this study; the authors used particle types and sizes available to them at the time of collection or purchase for the specific purpose of testing and validating the feasibility of using the SwisensPoleno instrument for the detection and

characterization of microplastic particles. An investigation on the detection efficiency of particles in this size class is needed and is reserved a future study.

Line 227-233: Dataset creation. Will you make this dataset available as an open data source, same goes with the model. Since this is a large aspect of the work I would expect to see this hosted somewhere, University server, Github or other so that the scientific community and others can make use of the data/model and build on it.

This was addressed above in a previous reviewer comment.

Line 261: Can you explicitly confirm here if CNN is used for the holo+fl or not.

The multi-input machine learning model using both the holographic images and relative fluorescence ("Holo.+Fl.") is a hybrid model type, utilizing components and architecture from models designed to classify particles using the holography ("Holo.-only", a convolutional neural network, or CNN) and the fluorescence ("Fl.-only", a multi-layer perceptron model, or MLP) separately. The combined model cannot be classified strictly as a CNN.

We have clarified the model structures in the manuscript by modifying text in Sections 2.4 (Lines 277, 279) as well as the description of the machine learning specifications in the Supplemental information (Section S2, Lines S32-S33) to state that the Holo.+Fl. model is a hybrid structure of a CNN and MLP.

Line 375-377: Could you expand your discussion to PA, PMMA and PA as they seem less distinct from the UMAP. Is there any further discussion that can be given to the important of dimensions 1 and 2 in the UMAP plot, are these evenly weight in terms of dimensionality? I am thinking along the lines of k-means plot dimension value assignment.

The dimensions produced using the Uniform Manifold Approximation and Projection (UMAP) visualization tool in the manuscript do not have weights unlike other clustering tools, such as k-means or principle component analysis (PCA). Dimensions 1 and 2 in Figure 6 of the manuscript instead represent coordinates in lower-dimensional space that preserve the input dataset's underlying topographical structure. They form a two-dimensional representation of the similarity or dissimilarity of the multi-dimensional information of the input data. In other words, the closer two points are to each other in Figure 6, the more similar their information; in contrast, the further away the points are from each other, the more dissimilar the data are. Thus, the multispectral fluorescence of PA, PMMA, and PE [the reviewer incorrectly referred to PA twice here] are more similar to each other than other particle types. However, the machine learning models that include fluorescence information as a model  The authors refer to the citing publication (McInnes et al., 2018) for additional information about the UMAP mechanics.

Line 387 – 500: Need to check classification values given and how stated, several read as incorrect or misleading, detailed in next comments.

Line 402: Should this be 78%?

No, as the volcanic ash dataset ("ash") had a classification accuracy of 0.8 (or 80%); thus, the statement ("...ash, mineral dust, hazel, PET, and PP particle types had an individual dataset classification accuracy less than 81%...") is correct as it stands.

Line 417: 93%?

The statement, "The accuracy for all pollen and MP particle types was greater than 92%..." is correct. The statement, "The accuracy for all pollen and MP particle types was greater or equal to 93%" is also correct, but we will leave it as it stands with the former statement.

Line 426 – 427: Please confirm the model used here.

Line 426 is the final line of the caption to Figure 8, which refers to Fl.-only model. Line 427 (of the unrevised manuscript) begins to report the classification accuracies of the Holo.+Fl. model, which is clearly stated on Line 428. Here are Lines 427-428 (of the unrevised manuscript): "The third model tested combined the holographic images and relative fluorescence approaches into a single, multi-input model ("Holo.+Fl.")."

Line 430: is this not less than 98% accuracy?

Yes, the authors agree with the reviewer that the classification accuracy of the volcanic ash and mineral dust datasets (85% and 82%, respectively) is less than 98% accuracy. This is clearly stated on line 430 of the unrevised manuscript: "An accuracy of less than 95% was observed only for the ash and mineral dust particle types (85% and 82%, respectively)."

An accuracy of 98% (Line 428 of the unrevised manuscript) refers to the overall accuracy of the entire model, considering all particle types. The range of classification accuracies for the Holo.+Fl. model for individual particle types is 0.72-1.00.

Line 449: related to line 402 accuracy reporting

The authors apologize to the reviewer for the confusion but stand by what is currently written on Lines 448-449 of the unrevised manuscript, which states: "...fragmented, irregular particle types in this study that had similar size distributions – such as PP, PET, volcanic ash, and mineral dust – performed with lower accuracies (accuracies < 81%)...". This statement is simply providing values of classification accuracies of the PP, PET, volcanic ash, and mineral dust datasets using the Holo.-Only model and the fact that they are less than 0.81. In other words, we are simply stating that the values 0.78, 0.69, 0.80,

and 0.59 (representing the classification accuracies of PP, PET, volcanic ash, and mineral dust, respectfully) are all less than 0.81.

Line 502-506: Should state for singular MP polymer groups? The sentences make it sound like you have measured all the particle types together which from my understanding is not the case.

The reviewer is correct that we did not measure all particle types together, at the same time, as was addressed previously. We have adjusted the text in the opening paragraph of the conclusion (Section 5, Lines 516-522) to emphasize and hopefully clarify that each dataset was measured separately:
"In this study, the high-performance capabilities of the SwisensPoleno's measurement system and application of a machine learning classification model were evaluated to accurately characterize and identify five different polymer types of MP particles under controlled laboratory conditions. The instrument's ability to identify and differentiate MPs from similarly featured coarse-mode aerosol particles, including mineral dust, various pollen taxa, and water droplets, was demonstrated. This was achieved through the application of a machine learning model that was trained and validated on separate datasets consisting of holographic images and fluorescence spectral data for each particle type. The high classification accuracy of the model affirmed the instrument's effectiveness in distinguishing between single coarse-mode particles."

Line 505: Can you claim that it is in real time classification, not an in real time measurement? My understanding is that individual particle types where measured. Later the analysis was made.

The reviewer is correct: for this study, the classification was performed after the datasets were created for the machine learning model, as stated in the instrument description (Section 2.1, Lines 131-132 of the unrevised manuscript): "For this study, machine learning classification models were created, trained, and evaluated in a separate Python programming environment decoupled from the instrument."

We have adjusted the opening paragraph of the conclusion (Section 5, Lines 516-522) based on the response to the previous comment, including the removal of the phrase "...near real-time...".

However, when the SwisensPoleno instrument is deployed to characterize and identify ambient particles, the classification happens in near real-time with a pre-trained model (such as the one developed for this study) initialized to classify particles during deployment.